# Mindfulness-Based Intervention Effects on EEG and Executive Functions: A Systematic Review

**DOI:** 10.3390/brainsci15030324

**Published:** 2025-03-20

**Authors:** Gilberto Galindo-Aldana, Luis Arturo Montoya-Rivera, Jose Jaime Esqueda-Elizondo, Everardo Inzunza-Gonzalez, Enrique Efren Garcia-Guerrero, Alfredo Padilla-Lopez, Tara G. Bautista, Cynthia Torres-González

**Affiliations:** 1Laboratory of Neuroscience and Cognition, Facultad de Ciencias Administrativas, Sociales e Ingeniería, Universidad Autonoma de Baja California, Carr. Est. No. 3 s/n Col. Gutierrez, Mexicali 21700, BC, Mexico; gilberto.galindo.aldana@uabc.edu.mx (G.G.-A.); arturo.montoya@uabc.edu.mx (L.A.M.-R.); 2Facultad de Ciencias Químicas e Ingeniería, Universidad Autonoma de Baja California, Calzada Universidad No. 14418, Tijuana 22424, BC, Mexico; jjesqueda@uabc.edu.mx; 3Facultad de Ingeniería, Arquitectura y Diseño, Universidad Autonoma de Baja California, Carretera Tijuana-Ensenada No. 3917, Ensenada 22860, BC, Mexico; einzunza@uabc.edu.mx (E.I.-G.); eegarcia@uabc.edu.mx (E.E.G.-G.); 4Facultad de Ciencias Humanas, Universidad Autonoma de Baja California, Mexicali 21700, BC, Mexico; alfredopadilla@uabc.edu.mx; 5College of Social and Behavioral Sciences, Northern Arizona University, Flagstaff, AZ 86001, USA; tara.bautista@nau.edu

**Keywords:** mindfulness, executive function, social cognition, EEG, frontal alpha

## Abstract

**Background.** Mindfulness-based interventions (MBIs) have emerged as an alternative intervention for symptoms of psychological and psychiatric conditions, such as depression, anxiety, and emotional discomfort. Over the last ten years, MBIs have established a growing body of evidence that shows cognitive and neurophysiological benefits. Depression and anxiety are conditions with a high prevalence in the world population. In developing countries, it is reported that, given the conditions of being at a social disadvantage, anxiety and depression are higher, resulting in compromised psychological well-being and mental health. **Objectives.** This systematic review aims to quantitatively and qualitatively assess changes in the neuropsychological, particularly executive functioning and social cognition domains, and electroencephalographical (EEG) effects of MBIs. **Methods.** A systematic review was conducted using the Preferred Reporting Items for Systematic Reviews and Meta-Analyses (PRISMA) in three databases, Web of Science, Scopus, and EBSCO MedLine complete; 14,464 articles were found, 141 articles evaluated the effects of MBI on executive functioning, and 16 included both as in qualitative and quantitative variables. **Results.** The qualitative results show that the research on the effects of MBI on behavior and cognitive skills, including executive function, social cognition, and EEG analysis, is very scarce but consistent in suggesting strong correlations on cognitive and electrophysiological alpha–beta proportions asymmetry on frontal areas. Undoubtedly, executive functions, as a behavioral regulatory and self-monitoring system, are the most popular study of interest in the literature, including emotional regulation, awareness, planning, social skills, and focused attention. Although there are fewer studies assessing the effects of MBIs on social cognition skills. The funnel plot showed a symmetrical distribution but ranked out of significant correlation. Most estimates of treatment effects are positive (58%); however, the average outcome observed did not significantly differ from zero. **Conclusions.** This study concludes that the research integrating the analysis of the electrophysiological and executive function effects of MBI shows important methodological variations and clinical conditions, which explains the significant results reported individually. Even when most of the literature reports positive effects of MBIs on several behavioral and neurophysiological domains, there are still confounding factors that must be taken into consideration by researchers and clinicians before attributing possible inaccurate or generalizable benefits.

## 1. Introduction

The high prevalence of symptoms of depression and anxiety deriving from multiple clinical conditions [1] in adults has led to the development of prevention techniques such as mindfulness-based intervention (MBI). This technique has demonstrated significant advantages, acceptance, and positive effects in reducing mood-related clinical symptoms [2]. MBIs are mainly based on two premises: the focus of attention and the absence of judgments about one’s own experiences. This technique has progressed through a series of conceptual definitions involving a self-oriented phenomenon, attention and awareness, external events, ethical mindedness, cultivation, and present-centeredness [3]. Mindfulness is widely incorporated into healthcare work training to reduce stress, promote compassion, and lower medical errors. Even to date, no scientifically operationalized definition of mindfulness has been accepted by all academic fields that carry out studies on the subject [4].

In addition to considering MBI as a preventive technique, they also help to promote the optimization of protective cognitive factors, such as attention skills, focused awareness, social skills, executive functions, self-regulation, and emotional control, among others. Mindfulness-based treatments have shown evidence of an increased improvement in overall quality of life, including the reduction in medical conditions such as migraines [5], diabetes [6], and cravings and their relationship with negative effects [7]. In particular, this technique has been recently shown to have positive correlations with psychological well-being and behavioral performance.

### 1.1. MBI’s Neurophysiological Correlations

Current studies demonstrate how brain neurons correlates and plasticity adapt to changes in consciousness. Expanding awareness during the mind–body response has neural correlates demonstrated through several neuroimaging techniques that show how the autonomic nervous system changes from a parasympathetic dominant state to a sympathetic dominant one. Connections with the heart and respiratory systems, as well as central nervous system structures, such as the thalamus and amygdala, and connectivity with cortical function, are identified [8].

Only a few studies have been able to characterize the neural basis of these effects. For example, previous findings [9] have hypothesized that high-mindfulness individuals would show patterns of brain activity related to a lower involvement of the default-mode neural network (DMN) at rest (reduced frontal gamma power) and a state of “task readiness” reflected in a more similar rest-to-task pattern compared to their low-mindfulness counterparts. Based on their results, the authors suggest that people who engage in mindfulness practices appear to have a specific electrophysiological pattern characteristic of reduced involvement of DMN and attentional processes. Similar results had been also replicated using mindfulness-based fMRI neurofeedback in adolescents with affective disorders [10], and exploring focused attention-voluntary control of mental content [11].

Electroencephalography studies the bioelectrical potentials from the brain’s synapse, which are recorded from the scalp and can be obtained invasively and non-invasively using computerized systems. The EEG devices have considerable capacity and a range of signal-processing software to record continuously. EEG devices might become increasingly integrated with other dynamic neuroimaging systems, including functional magnetic resonance imaging [12]. EEG exploration associated with the practice of MF has shown several advantages, generating evidence of neurophysiological effects. For example, studies have reported that when people have undergone dramatic experiences and practice MF, these experiences tend to end. Spectral analyses of EEG data around cessations of dramatic experiences showed [13] that these events were marked by a large-scale decrease in alpha power beginning about 40 s before onset and that this alpha power was lowest immediately after cessation. Furthermore, during the pre-cessation period, there were slight increases in theta power for the right, parietal, and central temporal regions of interest (ROI).

It is suggested that in conjunction with neuropsychological assessment, MBI practice through the development of controlled breathing increases alpha power during listening tasks and increases alpha rhythm suppression when detecting subsequent Stroop-type errors (which is a reliable index of self-monitoring assessment, in which the alpha rhythm is reduced, suggesting alertness to detected errors) [14]. This suggests greater error control in participants in the mindful breathing group. Other studies exploring the relationship between anxiety traits and MBI practice performance have identified anxiety as a predictive feature of observable EEG power, contributing to greater electrophysiological activation compared to what MBI contributes [15], suggesting a guide for future longitudinal studies on anxiety targeted with interventions such as mindfulness to characterize individuals based on their resting-state neurophysiology. In the case of studies with depressive conditions, the findings of the effect of MBI on brain electrical activity have shown controversial results. For example, Szumaska et al. [16] report that the MBI intervention technique reduces clinical symptoms of major depression. However, it does not affect frontal alpha rhythm asymmetry. However, a systematic review [17] reports that, by 2018, at least seven studies evaluating brain electrical activity in participants with major depression following MBI, in particular one of the studies mentioned above, suggested changes in frontal alpha rhythm asymmetry; another study by Keune [18], cited in the aforementioned systematic review, failed to demonstrate frontal alpha rhythm asymmetries. For his part, Schoenberg [19] examined the effects of MBI through theta power in front-medial regions, assuming that this is a reliable marker of depression.

EEG findings on MBI practice time have indicated that lower frontal gamma power at rest is associated with higher dispositional mindfulness scores. According to multivariate regression analysis, the global q-EEG index better explains the variations in dispositional mindfulness. Global q-EEG is a promising method for generating more detailed information about the neural effects of mindfulness in trained and dispositional programs [9]. Long-term MBI demonstrates that non-expert meditation practitioners report more frequent and profound episodes of mind wandering. In contrast, experienced practitioners report more frequent and deeper periods of persistent meditation, with corresponding EEG activations demonstrating increased frontal midline theta and somatosensory alpha rhythms during meditation compared to mind wandering in expert practitioners. This suggests an associated impact of decreased brain frequencies in mind wandering. The executive functioning, cognitive control, and active monitoring of sensory information are frequently related to frontal midline theta and somatosensory alpha rhythms. These bioelectrical patterns support the idea that similar brain processes may underpin the maintenance of internal and external attentional orientations [20].

The study of the alpha rhythm has directed research in further understanding how the brain organizes functional and bioelectrical activity when it is processing and adjusting emotional information. According to Allen and collaborators [21], research on frontal alpha modulation and its asymmetry dates back forty years, and it is currently a reliable marker for investigating mood conditions. Frontal EEG asymmetry has the benefit of being linked to a very effective conceptual model of motivation and emotion. According to the approach–withdrawal model of frontal asymmetry, a tendency to approach or engage a stimulus is indicated by increased relative left-frontal activity, while a tendency toward either increased withdrawal motivation or decreased approach motivation is indicated by decreased relative left-frontal activity [22]. From a neurological perspective, the frontal cortex in human beings sustain an interaction with limbic structures when evaluating and adjusting emotional reactions for a particular emotional behavior, where theta rhytm is suggested to be sensitive to emotion upregulation, whereas alpha seems to be not modified by emotion induction and regulation [23]. According to studies aiming to describe the role of theta (3 to 8 Hz) changes during emotional processing, the interaction between the prefrontal brain and the amygdala during emotion control processes is linked to oscillations in the theta band. More precisely, this suggests that, between three and five seconds following the start of regulation, frontal (Fz) theta oscillations increased during upregulation and downregulation via cognitive reappraisal of painful images [24].

### 1.2. Executive Function and Clinical Relevance Associations with MBI

Scientific evidence suggests that MBI can improve core aspects of executive function in clinical and non-clinical populations. The components most studied are inhibitory control, working memory, and cognitive flexibility. These benefits also extend to attention, cognitive control, and emotional regulation, essential for adaptive functioning. At the neurobiological level, the reported changes include increased activity in the prefrontal cortex, anterior cingulate, and autonomic nervous system, key structures for emotional modulation, and cognitive control [25].

Studies such as Khatri et al. [26] demonstrated that even brief interventions can improve performance on executive functioning tasks. Their research with healthy adults found that combining physical exercise with mindfulness modulated cognitive flexibility and improved working memory.

On the other hand, in non-clinical populations, such as college students, individual factors such as trait mindfulness or integrative self-understanding have been observed to mediate the effects of MBI, suggesting that participants’ baseline characteristics significantly influence outcomes [27,28].

A pilot study evaluated the immediate effects of a 13 min mindfulness meditation (MM) session on 24 college students with no prior meditation experience. The results showed that meditation significantly increased mindfulness and reduced anxiety in all participants. Students with higher mindfulness traits showed higher levels of state mindfulness before and after meditation. Furthermore, better baseline performance in mindfulness was correlated with more significant increases in body awareness after practice, although no significant correlations were found with executive functions. The results suggest that a brief meditation session may be helpful for stress management in students, and individual differences in attention and mindfulness may influence the benefits obtained [28].

Meanwhile, the study by Bolzer et al. [29] explores the role of mindfulness, defined as mindful and deep cognitive processing of feedback, as a key to its effectiveness. Using eye tracking, they assessed fixations and visual transitions between a text and feedback comments (with and without justifications) in 30 students. The results showed that participants without justifications made more transitions and longer fixations, indicating a more significant integrative effort to relate the feedback to the text. Mindfulness was positively associated with feedback recall but not with text review performance, suggesting that its impact on performance depends on additional factors such as cognitive load. This finding underscores the importance of conscious processing in improving retention and effective use of feedback. Social cognition has also benefited from mindfulness-based interventions, particularly in areas such as empathy, theory of mind, and emotional recognition. Campos et al. [30] compared meditators and non-meditators and found that meditators demonstrated greater empathy, theory of mind skills, and lower hostile attributional style. These findings highlight the importance of non-reactivity to internal experience as a mediator of performance on social cognitive tasks.

MBI are feasible and practical in clinical populations, such as individuals with psychotic spectrum disorders. Mediavilla et al. [31] implemented a pilot study with patients diagnosed with schizophrenia, finding significant improvements in the theory of mind and emotional recognition. Likewise, Lopez del Hoyo et al. [32] identified significant differences between clinical groups and healthy controls in social cognition and mindfulness skills, highlighting the latter’s role as a mechanism that promotes the appropriate interpretation of mental states and emotions of others.

In the case of people with autism spectrum disorders (ASDs), Simione et al. [33] conducted a systematic review of 37 studies. Their findings revealed improved social skills, stress reduction, and psychological problems. However, they also highlighted the need to adapt interventions to the heterogeneity of this population. Liang and Shek [34] conducted a randomized control trial with people with physical disabilities, showing that MBI improved levels of mindfulness, self-compassion, and perceived well-being, along with a significant decrease in anxiety and depression. This suggests that interventions may be helpful tools to improve the quality of life in populations with physical limitations. Although MBI has shown promising results in different populations, there are some limitations to generalizing the results. The heterogeneity of clinical conditions, methodological designs, and the influence of contextual variables such as educational level and individual characteristics of participants are relevant factors to consider when designing and implementing these MBIs. In non-clinical populations, Ede et al. [35] showed that mindfulness as a trait is related to lower perceived stress and cardiovascular reactivity. However, its effects on physiological variables require active conscious states and prolonged practices. Mindfulness-based interventions are promising for improving executive functions and social cognition in clinical and non-clinical populations. The findings suggest that their efficacy is mediated by neurobiological and psychological changes and participants’ characteristics. However, further research is still needed to fully understand the mechanisms behind these effects and how to optimize these interventions for different population groups.

There is also a large body of literature demonstrating the usefulness and viability of mindfulness interventions for individuals with autism spectrum disorder (ASD) [33]. The review studies show positive results, including a decrease in behavioral problems, a reduction in psychological distress, and improvements in social and cognitive skills in people with ASD. Even with these encouraging outcomes, the study highlights the lack of MBI for pediatric patients and advises care when approving the body of research. The findings also highlight how urgent it is to investigate customized interventions for various ASD subgroups, taking into account the spectrum of autism and increasing support for educators in learning environments.

Increased rates of attention deficit hyperactivity disorder, depression, drug abuse, and antisocial behavior are just a few of the short- and long-term issues that have been linked to poor executive function. It has been demonstrated that MBI that emphasize raising awareness of one’s thoughts, feelings, and behaviors enhance particular EF traits like emotion regulation, cognitive control, and attention. Randomized control trials of MBI show improvements in particular EF components and specifically point to the autonomic nervous system and the anterior cingulate cortex’s neural circuitry as two brain-based processes behind improvements associated with MBI [25].

Concerning social skills, mindfulness benefits social cognition abilities [30] and has demonstrated improvement in clinical psychological and psychiatric conditions. For example, the literature presents the advantages of the People With Psychosis Improve Affective Social Cognition and Self-Care After a Mindfulness-Based Social Cognition Training Program (SocialMIND) [36] for people with psychosis [31,37]. Social cognition and mindfulness may even be a useful practice in individuals with schizophrenia or obsessive–compulsive disorder [32], as well as affective symptoms in persons with schizophrenia [38].

However, little is known about mindfulness-based interventions’ effects on non-clinically significant symptoms or may improve other social cognitive skills, such as empathy, or attentional processes for face recognition [39], and face–composite construction process to facilitate identification [40,41]. Accordingly, meditation is promising for integrative self-knowledge and well-being [27].

Other electrophysiological measures had been ERP-oriented, involving inhibitory, and change detection control, [42,43], and proposed that the effects of MBI produces less error rates during the resolution of inhibitory control tasks, accompanied by an increase in NoGo-P3 amplitude when compared to control peers.

Research on MBI’s advantages in well-being, cognitive, and physiological domains has earned scientific importance, and the different methodologies implemented to test the hypotheses about the benefits of the treatments have demonstrated reliability and a low risk of bias. However, for the past decade, some systematic reviews and meta-analyses have been published reporting the effects of MBI on different cognitive and behavioral domains, for example, emotional intelligence [44], post-traumatic stress disorder [45], distress [46], burnout [47], elderly [48], pain or physical conditions [49,50,51,52], among others. On the other hand, only a few had informed about neurophysiological effects, for example, ERP [53], EEG and fMRI [17,54,55], or neuropsychological domains, for example, executive functions [56], attention [57,58,59]. However, to the best of our knowledge, no reviews integrate these variables and the effects of MBI on social cognition skills. The lack of integration of these variables as a whole in the same study may lead to limitations or erroneous generalization and statistical power of the findings due to the diversity of research conditions, such as the age of the participants, health status, or culture, and research models, such as clinical trials or experimental. Therefore, the present systematic review aims to describe the qualitative and quantitative properties of published research from the past 10 years that have reported MBI effects on executive functions and electroencephalography.

The rest of the manuscript is organized as follows: Section 2 provides the methodology, considering the search strategy, search string, selection process, data analysis, qualitative synthesis, significant outcomes, minor outcomes, condition of the studies, quantitative analysis, and risk of bias. Section 3 shows the findings about the qualitative synthesis, quantitative analysis, and risk of bias assessment of the included articles. Section 4 presents the discussion according to the main results are presented.

## 2. Methodology

### 2.1. Search Strategy

Due to the completeness of the discipline’s publications, EBSCO, Web of Science, and Scopus were the selected databases for this analysis.

### 2.2. Search String, Selection Process, and Data Analysis

The following variables were considered for the search: EEG OR Electroencephalography AND Mindfulness AND executive function OR Social Cognition. The search was limited to 10 years (2014–2024) and academic peer-reviewed publications, and the following strings were used: (((((((TS = (“EEG”)) OR TS = (“Electroencephalography”)) AND TI = (“mindfulness”)) AND TI = (“executive function”)) OR TI = (“Social Cognition”))) NOT TI = (ERP OR “event related potentials”)). For EBSCO, MEDLINE complete, 376 documents were retrieved. For Clarivate (((((((TS = (“EEG”)) OR TS = (“Electroencephalography”)) AND TS = (“mindfulness”)) AND TS = (“executive function”)) OR TS = (“Social Cognition”))) NOT TS = (ERP OR “event related potentials”)) 13,147 documents. For Scopus EEG, Electroencephalography, Mindfulness, Executive Function, and Social Cognition, 12 documents are shown. Reduced variables search included: (((“EEG”)) AND (“Mindfulness”)) AND (“Executive Function”))), giving a total of 909 articles, Clarivate showing 20 results. The Population Intervention and Outcome facet (PICO) filter was applied to automatically remove wrong outcomes, animal and cellular studies, systematic and literature reviews, meta-analyses, and non-intervention studies.

Automated tools removed systematic reviews, meta-analyses, and animal studies, using PICO (Population, Intervention, Comparison, and Outcome) intervention. Three authors from this study decided to include using Rayyan AI for blind review and inclusion decisions. Of the 141 reports assessed for eligibility, 38 were excluded because they did not include the variables of interest for the present review, 15 studies were not in human samples, and 72 did not present an intervention report. A total of 27 conflict decisions were resolved in a second iteration by applying exhaustive inclusion criteria to the studies. Seven agreements were acquired after the first Blind revision, a second blind-off revision was made, resulting in 16 studies to be included (Figure 1).

### 2.3. Qualitative Synthesis

Data from the included research are charted in Table 1, including year, authors, country, participants’ characteristics (sociodemographics), sample size, intervention program features, types of intervention, EEG procedures (resting state or task-related), outcomes, executive function assessment procedures (follow-up or pre-/post-intervention), gender, outcomes, and statistical inference report from research results.

### 2.4. Major Outcomes

#### 2.4.1. Bandwidth, Region of Interest, and Filtering Techniques

Theta (4–<8 Hz), alpha (8–<12 Hz), and beta (12–<30 Hz), relative power was calculated for each electrode and log-transformed. Signals were recorded using a bandpass filter of 0.5–40 Hz and sampled at a rate of 500 Hz. Channels with excessive artifacts were interpolated [60], re-referenced offline to the averaged mastoid references, and bandpass filtered from 1 Hz to 30 Hz, with eye movements and blink artifacts corrected by independent component analysis (ICA) [61]. Individual frequency bands, individual alpha frequency, filtered at 0.1–200 Hz (zero-phase; roll-off: 12 dB LP/24 dB HP) and referenced to the average of all remaining channels, and neural from non-neural signal sources by second-order blind source identification [62,63,64], alpha1, 6.9–8.9 Hz; alpha2, 8.9–10.9 Hz; alpha3, 10.9–12.9 Hz [65]. The power of various brain rhythms, such as delta (0.5–4 Hz), theta (4–8 Hz), alpha (8–13 Hz), and beta (13–30 Hz) bands, estimated using fast Fourier transformation on EEG [64]. Mean EEG data filtered using hamming windowed FIR filter with a passband of 0.1 to 45 Hz, independent components associated with eye movement and muscle activity visually examinated, and eliminated using ICA [66]. Vertical eye movements were identified through the use of ICA and subsequently removed, with the impedance level kept below 20 kΩ, as well as the notch filter (to 60 Hz) [67].

#### 2.4.2. Mean Power Spectral Densities (μV2/Hz)

Fast Fourier transform (FFT) [14,68], natural logarithm transformation for the theta (4–8 Hz) and alpha (8–12 Hz) bands [69], power spectra and multitapered density [63], absolute power [70], short-time FFT peak detectability estimation (find peak approach), time-locked EEG epochs band-pass filtered offline (0.5–30 Hz), and Fast Fourier Transform were used to decompose samples into theta (5–7 Hz), alpha (8–14 Hz), and beta (15–29 Hz) frequencies [67].

#### 2.4.3. Alpha Asymmetry

Frontal alpha asymmetry was determined by the subtraction of the log-transformed left hemispheric site values from log-transformed values from the right hemisphere, signal-to-noise ratio as the ratio of the absolute power in a frequency bin divided by the surrounding ±5 Hz, and excluding the surrounding ±1 Hz [71]. It had, for each electrode, average alpha power densities [72]; alpha under the concentration task [73]; and relative power coherence and symmetry [61,74].

#### 2.4.4. ERP N200 and P300

Fronto-central region No-Go P300 and N200 was used for examining the executive resolution of conflict through inhibitory control [75].

#### 2.4.5. Inhibitory Control and Emotional Regulation

For inhibitory control and emotional regulation, the following were used: self-assessment manikin [76], emotional valence ratings [71], emotion recognition, go/no-go task measures [75], Location–Direction Stroop-Like Arrow Test, number of correct responses and reaction time, Hearts and Flowers Test reaction time of correct answers and accuracy (number of errors in congruent and incongruent trials), Adapted Stop Signal Task (SST) [70], Stroop test, and time [14,65,77].

#### 2.4.6. Planning

The Trail Making Test was used to determine the time to task completion [65,70,72,78].

#### 2.4.7. Memory

For memory, the following were used: for working memory measures, the operation span task and number of correct items in correct position [64,79,80]; for verbal memory, California Verbal Learning Test (CVLT) [65,81]; as well as visual stimuli reaction time [66,82].

#### 2.4.8. Attention and Awareness

For sustained attention [63,83], the Sustained Attention to Learning scale was used to examine the following parameters: self-regulation, sustained attention, text messaging during class, and cognitive learning [64,84]. Moreover, we also used attention and attention switching, Attentional Matrices [65,85], Breath focus task, self-assessment manikin, level of arousal and emotional valence [67,76], and the Chinese Affective Picture System [61,86].

### 2.5. Minor Outcomes

#### Mood

The following were used to determine mood: State-Trait Anxiety Inventory, Positive and Negative Affect Schedule (PANAS) [69], Center for Epidemiologic Studies Depression Scale (CES-D) [72,87]; Feeling scale [68,88], affective arousal, Felt Arousal Scale [68,89], depression, anxiety, and stress state (DASS) [73,90], Langer Mindfulness Scale (LMS) [64], Five-Facet Mindfulness Questionnaire (FFMQ) [91], and trait anxiety from Penn State Worry Questionnaire (PSWQ) [14,92].

### 2.6. Condition of the Studies

#### 2.6.1. Participants

The study comprised human participants, including young and neurotypical adults, for the control groups. For the intervention groups, mental health-related conditions include anxiety, depression, dysexecutive functioning, and/or social cognition impairment. The age of the sample was also considered. The data extracted from the literature were charted by continuous discussion by the collaborators of this research to obtain a final iteration of the results from the included reports.

#### 2.6.2. Intervention

MBIs were sought. The reports aimed include objectives, the number of sessions, and a clear description of how at least one of the interventions follows mindfulness techniques.

#### 2.6.3. Comparator

The comparative results from studies were mainly psychoeducation, cognitive therapy, or waiting list groups as control comparisons.

### 2.7. Quantitative Analysis and Risk of Bias

The quantitative analysis was performed by obtaining EEG features results, and neuropsychological outcomes from before and after MBI or comparison groups were collected from 11 reports. For its inclusion, the report should inform similar methods for measuring electrophysiological effects of MBI. It should have one of the following conditions: a control group, the same-sample-size group in a wait list or placebo intervention, or a pre-/post-mindfulness intervention measure. For each study, the number of participants were calculated as means. Standard deviations per group or condition were extracted from the intervention condition of interest to calculate raw difference statistics—pooled standard deviations, significance *p* values, mean differences, and confidence intervals—and standardized effect size—effect size, bias-corrected, standard error, and confidence intervals. Publication bias was assessed using funnel plot asymmetry and Egger’s regression test. To estimate the meta-effect from standardized mean differences, we applied Hedge’s *g* standard error of the observed outcomes as a moderator.

**Table 1 brainsci-15-00324-t001:** Qualitative analysis of neuropsychological and electrophysiological effects of MBI of the included articles.

EEG Outcome	Measure Condition	Brain Region	Neuropsychological Outcome	n	Design	Intervention Types and Gender	Sample Condition	Ref.
Bandwidth and Region of Interest. Treatment-dependent reduction in beta power	Resting state, eyes closed	Left and right frontal and central	Unclear outcome	57	Clinical trial	Cognitive therapy, mindfulness meditation, mindfulness-based cognitive therapy, 52% female, 48% male	Chronic low back pain	[60]
Alpha asymmetry. Treatment dependent leftward shifting	Visual emotional stimuli reaction pre-/post-training	Dorsolateral frontal (F7, F8)	Emotional regulation	67	Randomized Clinical trials	Focused attention 19% female, open monitoring, 27% female, mindfulness-based cognitive therapy, 21% female	Emotionally evocative visual stimuli	[71]
FFT. No differences in alpha, and higher theta power between intervention types	During 30 min of intervention	Frontal	Emotional regulation to negative state	35	within-subject crossover	Mindfulness induction, relaxation induction, 27 male, 8 female,	State anxiety and negative affect	[69]
Average of alpha power densities. Stable left frontal alpha	Resting state, counterbalanced closed–open eyes	Frontal (F3, F4)	Increased functional attention and awareness in tests of planning performance, improvement of attention and executive control	110	Randomized control trial	Mindfulness-based stress reduction, 62% female, and waiting list control 62%	Perceived stress and depressive symptoms	[72]
Conflict monitoring N200 and P300	Conflict monitoring Go/No-go	Frontal central (FCz)	Improvements in dynamically deployed cognitive control in processing and responding to facial expressions of emotion	66	Randomized controlled study	Brief mindfulness training intervention and book learning, control 31% male, 69% female	Intervention credibility and expectancy	[75]
FFT. Enhanced beta coherence between right frontal and right temporo-parietal	Focusing on past and future thoughts, during auditory paced walking	Frontal dorsolateral (AF7, AF8) and temporo-parietal (TP9, TP10)	Attention, focal awareness	24	Clinical trial	Mindfulness-mindlessness meditation and control condition, 12 male, 12 female	Physical task	[68]
FFT. Cortical alpha activity	Calm condition	Prefrontal (AF3)	Improvement in emotional regulation and concentration	25	Longitudinal	Mindfulness-based stress reduction training course, 9 male, 16 female	Depression, anxiety, stress state, electrodermal activity	[73]
Relative power. Alpha increase	Biofeedback during mindful meditation	Not reported	Problem solving skills improvement, unclear outcome	40	Controlled trial	Mindful meditation, biofeedback, gender not specified	Adolescence, Stress management, Culture (urban–rural)	[74]
Power spectra. Individual alpha frequency. Beta band reduction, training related reduction of individual alpha frequency	Mindfulness of breathing meditation	Anterior, and posterior central	Focused attention	60	Controlled trial	Meditation techniques, wait list, gender matched 50%	Time of meditation practice	[63]
Individual bandwidth. Reduction in the theta/beta ratio	Working memory automated operation span task	Frontal, parietal, occipital, and central	Working memory, metacognition, social skills	24	Randomized control trial	Mindfulness training, all female students	Academic achievement	[64]
Intervention group showed no change in alpha and theta absolute power, compared to control group showing a significant decrease	Rest with eyes-open condition	AF7, AF8, TP9, and TP10	Inhibition and flexibility improvement, decrease in reaction times	22	Exploratory pilot study, randomized controlled trial	Mindfulness training, EEG-feedback, sex not reported	Healthy children	[70]
Individual alpha frequency. Alpha 2 increase, alpha 1 decrease	Resting state	Not reported	Improvement in verbal memory, attention switching and executive functions, interoceptive awareness, and rumination	50	Clinical trial	Web-based MBI, 74% female	Older adults, COVID-19	[65]
Individual alpha–theta amplitudes. Short-time FFT. Mind wandering is associated with an increase in the amplitude and a decrease in theta frequency. Alpha showed a decrease in amplitude and increase in frequency during mind wandering relative to breathing focus	Condition-related	Several scalp areas	Awareness	28	Experimental	Focus on the sensation of breathing, Mind wandering 11 males, 17 females	Novice meditation practitioners	[67]
Alpha bandwidth power asymmetry scores. Greater left, relative to right, asymmetry than low mindfulness trait during emotion regulation	Emotional stimuli reaction	Left and right frontal	Cognitive affective arousal earlier recognition of emotional stimuli	92	Experimental	High and low Mindfulness trait, 49 females and 43 males	Adolescence	[61]
Individual bandwidth. Change in band powers while handling cognitive load difference utilizing cognitive ability modulation index (CAMI) feature	Pre- and post-MBI	FP1, FP2, AF3, Aand AF4	CAMI improvement after MBI	40	Clinical trial	MBI, and control group, sex not reported	Healthy young adults	[82]
FFT. MBI-induced changes in alpha reactivity during response-monitoring	Error detection task	F3, Fz, F4, C3, Cz, C4, P3, Pz, P4	No treatment condition effects on attentional control, accuracy, and reaction time for conflict detection	65	Clinical trial	Mindfulness, 30 female, 14 male, and control condition exercise, 16 female, 5 male	Healthy young adults	[14]

### 2.8. Protocol Registration

This systematic review protocol was registered in the National Institute for Health Research (NIHR) International Prospective Register of Systematic Reviews (PROSPERO) under the registration code: CRD42024531251.

## 3. Results

### 3.1. Qualitative Synthesis

The literature addresses a spectrum of several mood and medical conditions. For instance, many of these conditions often co-occur, presenting physical diseases accompanied by mood symptoms or even disorders. It is well known that psychological or psychiatric conditions may require particular treatments depending on factors such as cognitive reservoir, attention conditions, the severity of the condition, or the need for pharmacological interventions.

Figure 2 illustrates the reported significant associations between behavioral and neuropsychological functions and MBI-dependent changes registered on the EEG scalp source in three main frequency bands: beta, alpha, and theta. Of the articles included in this study, eight fully reported a specific electrode EEG activity change related to implementing an MBI. According to these studies, neuropsychological functions such as planning, executive control, emotional regulation, focused attention, awareness, working memory, meta-cognition, and social skills are shown to improve as a result of MBI, compared to control or pre-intervention groups.

The neurophysiological effects of mindfulness meditation, cognitive therapy, and mindfulness-based cognitive therapy have been studied using EEG brain oscillation changes [60]. Day and collaborators identified particular EEG features associated with outcomes across treatments in their report. In their sample, a group of participants with chronic low back pain conditions was associated with increased cortical arousal as expressed by fast beta activity and a reduction in alpha and theta oscillations related to a reduction in beta power. Similar decreases in beta power were observed in all five regions of interest (frontal left, frontal right, central left, central right, and central) when implementing mindfulness-based cognitive therapy (MBCT) due to the long time impact. Research on the effects of MBCT has previously demonstrated cognitive and neurophysiological effects, described in terms of reactivity to visual, emotional stimuli, showing consistent leftward shifts in focused attention group, before and after the MBI training procedure [71], the authors refer to a main emotional-related change in F8-F7 electrodes across all three training sessions, and frontal alpha asymmetry activation became more left-sided.

In older adults, MBI showed an increase in planning skills. Clinical trials comparing the effects of mindfulness-based stress reduction (MBSR) with a wait-list control group showed changes in left frontal alpha asymmetry, suggesting an improvement in positive emotion processing and an improvement in adaptive immunity measured with immunoglobulin G response to a protein agent. In addition, MBSR showed a critical change in executive functioning planning skills after 8 weeks of intervention, observed in a reduction in the time to execution ratio on the Trail Making Test [72]. The EEG changes reported showed a moderately significant interaction between treatment group and time when these data were analyzed for the initial time and time 2 after the intervention. The interaction represented moderate differences between treatment groups at time 2. The F3 and F4 asymmetry did not differ between the MBSR and waiting list groups at time 1 or after times assessments. However, MBSR individuals showed significantly more significant leftward alpha asymmetry than wait-list subjects immediately after completing the 8-week MBSR program. In particular, MBSR patients maintained a consistent alpha asymmetry between baseline and treatment completion. These results are in contrast to the significant increases in frontal left-sided alpha asymmetry observed in a different research from Saggar et al. [63].

Compared to the control condition, an MBI group and relaxation induction similarly decreased state anxiety and negative affect from the pre-test to the post-test, according to repeated measures analysis of variance. The three situations did not differ significantly regarding positive impact over time. While there were no differences in alpha power between treatments, individuals’ frontal theta power was higher during MI and RI than in the control condition. Through comparable brain processes, the current study offers preliminary electrophysiological evidence that brief MI and RI alleviate negative psychological states in individual sports athletes. Future research must fully examine the moderating effects of training experiences and long-term therapies on athletes’ mental states and EEG activity [69].

There is a large body of literature related to the effects of MBI on emotional states [93,94], even online-administered programs [95]. However, there is also evidence of effects on precursor elements for executive functions of greater complexity in cognitive processing. For example, concerning the effects of MBI on social information processing to promote higher socioemotional skills, the evidence suggests that MBI may improve stimulus discrimination scores by enhancing facial expression discrimination. In a study from Quaglia and collaborators [75], individuals with lower baseline discriminating scores showed the most substantial effect of MBI, indicating that mindful attention training would be more advantageous for individuals with more need for adjustment of this skill. The study suggests an enhancing effect on executive functions such as speed and accuracy of behavioral performance on the emotional go/no-go task—which includes keeping a task-relevant goal in mind, choosing goal-relevant information, inhibitory control, and continuous performance monitoring.

Studies of physical activity and its correlation with EEG and cognitive characteristics suggest a particular approach to understanding MBI effects. MBI psychological effects of meditating during periods of mild physical activity may be linked to improved interhemispheric connectivity between beta range’s left temporo-parietal and right frontal areas [68]. Continuously shifting the focus to the here and now can improve affective states, a perceived down-modulation activation more than usual, and an increased awareness of thoughts, feelings, and bodily sensations. In their study, Bigliassi et al., using a portable EEG system, measured AF7, AF8, TP9, and TP10 scalp derivations. They found beta magnitude-squared coherence values differing between a control condition and a mindfulness condition, and this coherence increased when participants performed a physical task. Their findings address that, behaviorally, MF may promote the usage of associated thoughts, potentially blocking the entry of task-unrelated thoughts into focal awareness, and electrophysiologically, would increase functional connectivity between frontal and dorsal brain regions related to interoceptive awareness.

The efficiency of MBI may be measured across sessions utilizing alpha modifications. The alpha power spectrum in frontal regions seems to present changes after the third week of an MBI. According to Morais et al., both a subset of course participants initially assessed as less healthy, and the overall change in alpha activity present significant differences. Participants with higher anxiety scores and those in better health appeared to be less affected by this power shift. The overall p-values of the EEG data and the heart rate measures are reasonably comparable. The rise in alpha power is noticeable throughout the first few weeks of the MBI course and becomes even more pronounced after the third session when it increases to almost 250% of the initial values [73].

In other studies, regulatory properties related to the involvement of subcortical brain structures have been emphasized, including amygdala, and anterior cyngulate cortex (ACC), as bottom-up cognitive control regulators from exterior sensory stimuli. Subcortical structures such as the amygdala are essential in processes such as social cognition [96]. However, it also contributes to regulating the effects of external world stimuli in behavior. In their study, Hölzel et al. [97] found, after both MBI and a stress management education program, a statistically significant pre/post decrease in right amygdala activation in response to neutral faces but no significant differences between the therapies and no discernible changes in amygdala activation in response to joyful or angry faces after either intervention. From a cortical–subcortical point of view [98] Nakamura et al. suggested that the influences of the MBI brain circuit mainly constituted top-down emotion regulation structures including premotor frontal areas, orbitofrontal, and dorsolateral cortex, ACC, and insula, inhibiting amygdala sensory signals.

Differences in emotion control techniques during mindfulness may result from changes in the distribution of processing resources allocated to brain activity brought about by mindfulness meditation practices. Additionally, considering the variations in training phases and duration of the mindfulness meditation experience, Nakamura’s simulations sought to clarify the information propagation process in the brain during mindfulness.

Also, individual differences such as culture and development may play a relevant role in problem-solving skills, stress management, and the ability to use EEG biofeedback while practicing mindful meditation [74].

According to Saggar et al. [63], the intensive training of focused attention skills for long periods of time can produce changes in the brain’s electrical activity during meditation, suggesting possible long-term improvements in cognitive processes in general. Even MBI training has been implemented in combination with EEG feedback, suggesting promising benefits on executive functioning and EEG power effects [70]. In their study, Vekety and collaborators evaluated executive functions, including inhibition, flexibility, and planning, in two groups: mindfulness with EEG feedback and a control group. The first statistical model from the authors suggests no positive effects on the evaluated executive functions and EEG. However, a second iteration of analysis suggest differences between groups before and after the intervention.

In the case of the study by Bing [14], two different EEG analyses were performed. From one side, in alpha power calculation during an audio exercise and during a Stroop task, the electrode source varied; for analyzing stimulus processing in the Stroop task, changes in alpha power were extracted from F3, Fz, F4, C3, Cz, C4, P3, Pz, and P4. The sources were obtained from the midline electrodes Fz, FCz, Cz, and Pz for the analysis of ERPs and error-related negativity.

### 3.2. Quantitative Analysis

The analysis used the standardized mean difference as the outcome measure. A random-effects model was fitted to the data. The amount of heterogeneity (i.e., tau²) was estimated using the restricted maximum-likelihood estimator [99]. In addition to the estimate of tau², the Q-test for heterogeneity [100] and the I² statistic are reported. If any amount of heterogeneity is detected (i.e., tau² > 0, regardless of the results of the Q-test), a prediction interval for the true outcomes is also provided. Standardized residuals and Cook’s distances are used to examine whether studies may be outliers and/or influential in the model context. Studies with a standardized residual larger than the 100 × (1 − 0.05/(2 × k))th percentile of a standard normal distribution are considered potential outliers (i.e., using a Bonferroni correction with two-sided alpha = 0.05 for the studies included in the meta-analysis). Studies with a Cook’s distance larger than the median plus six times the interquartile range of the Cook’s distances are considered influential. The rank correlation test and the regression test, using the standard error of the observed outcomes as a predictor, are used to check for funnel plot asymmetry.

A total of 21 outcomes were included in the analysis. The observed standardized mean differences ranged from −9.1655 to 4.6441, with most estimates being positive (57%). The estimated average standardized mean difference based on the random-effects model was $\hat{\mu }$ = −0.4056 (95% CI: −1.4838 to 0.6725). Therefore, the average outcome did not differ significantly from zero (*z* = −0.7374, *p* = 0.4609), as shown in Figure 3.

### 3.3. Risk of Bias Assessment of the Included Articles

According to the Q-test, the true outcomes appear to be heterogeneous (Q(20) = 552.7606, *p* < 0.0001, tau² = 6.2465, I² = 98.9894%). A 95% prediction interval for the true outcomes is given by −5.4214 to 4.6102. Hence, although the average outcome is estimated to be negative, in some studies, the true outcome may be positive. An examination of the standardized residuals revealed that one study [65] had a value larger than ±3.0381 and may be a potential outlier in the context of this model. According to Cook’s distances, several studies could be considered overly influential. The regression test indicated funnel plot asymmetry (*p* < 0.001) but not the rank correlation test (*p* = 0.353), as presented in Figure 4.

## 4. Discussion

This systematic review aimed to describe the qualitative and quantitative properties of published research from the past 10 years reporting MBI effects on executive functions and electroencephalography. According to the findings, the main correlations to describe cognitive and electrophysiological effects of MBI suggest a wider number of studies related to the frontal brain areas [72]. However, some studies deliver a broader understanding of EEG modifications and neuropsychological functions by approaching theta–beta ratios instead of alpha asymmetry [64]. Undoubtedly, executive functioning as a behavioral regulatory and self-monitoring system is the most frequent interest of study in the literature. Still, fewer studies are related to social cognition effects measuring on this domain.

In particular, a study [73] has developed a model involving several psychological, neurophysiological, and psychophysiological variables during a period of four evaluation sessions. This may lead to the observed differences compared to other similar studies.

Another study showing a large outlier compared with the studies included in this report is that from the data of Rastogi et al. [74]. This study presents a particular approach in the discipline, and the authors implemented a neurofeedback technique combined with mindful meditation. Newer methods of monitoring may require greater sensitivity of efficiency but, at the same time, offer better clinical outcomes for participants. The effects could even be related to the techniques implemented for the administration of MBI, since, for example, in Galluzzi’s [65] study, standard differences were also observed in the analysis of alpha power.

Even when a large body of the literature reports the positive effects of MBI on multiple behavioral and neurophysiological domains, researchers and clinicians must consider that there are still confounding factors before attributing possible spurious benefits. Sorjonen and Melin [101] suggest that the meta-analytic cross-lagged effects, when calculated after correcting for a previous outcome variable measurement, could still be erroneous because of regression to the mean and correlations with residuals. After fitting competing models to simulations of the same meta-analytic data, they concluded that the mindfulness components’ potential effects on anxiety and depression symptoms were most likely fictitious. More research is needed to define particular effects among specific behavioral features through clinical trials. The findings in state mindfulness may be a more realistic predicted outcome. Still, the data from current research support the use of mindfulness induction to reduce state anxiety in anxious persons. However, to distinguish the relative impacts of objectively measured anxiety outcomes from mindfulness induction in clinically characterized samples, further controlled trials are required [94].

The level of meditation skill provides evidence of variation in the neurophysiological and cognitive effects of practicing MBI, which could be a confounder in several studies. Meditation training reduces susceptibility to mind-wandering, and long-term meditation practice cultivates prolonged and internally directed attentional states of awareness, as evidenced by enhanced theta activity over mid-frontal theta areas and alpha activity primarily concentrated over the somatosensory cortex, according to evidence from [20].

Another identified factor observed in the literature included in this systematic review relies on the heterogeneity of measures regarding the gender of the samples in the studies, making it difficult to make direct comparisons for all samples. At least eight of the studies report more than 50% of female sample in the groups. According to the literature, sex is an independent variable that should be carefully considered when the alpha rhythm is part of the research interest, once the menstrual cycle may impact the alpha reactivity to open and close eye maneuver [102], as well as to cognitive neuropsychological domains [103].

According to the number of articles identified in this systematic review, there is a lack of reviews that include social skills. The literature addresses the benefits of MBI for clinical psychiatric conditions but lacks research with non-clinical samples, such as adolescents or young adults with neurotypical development.

## 5. Limitations and Future Directions

This systematic review has several limitations; for example, we did not analyze the technical conditions of the EEG measures reported in the included articles. In recent years, low-cost devices have improved their efficiency, reducing artifacts and increasing the sensitivity and resolution of the information obtained. However, there are still aspects related to the interpretation of the signal source for the algorithmic attribution of its origin that require greater control and verification by means of gold-standard tests of clinical range equipment. Another important issue for this study is related to the complexity and heterogeneity of the samples from the included research, for example, sex and culture, making it necessary to be considered as independent results and making it difficult to perform direct comparisons.

## Figures and Tables

**Figure 1 brainsci-15-00324-f001:**
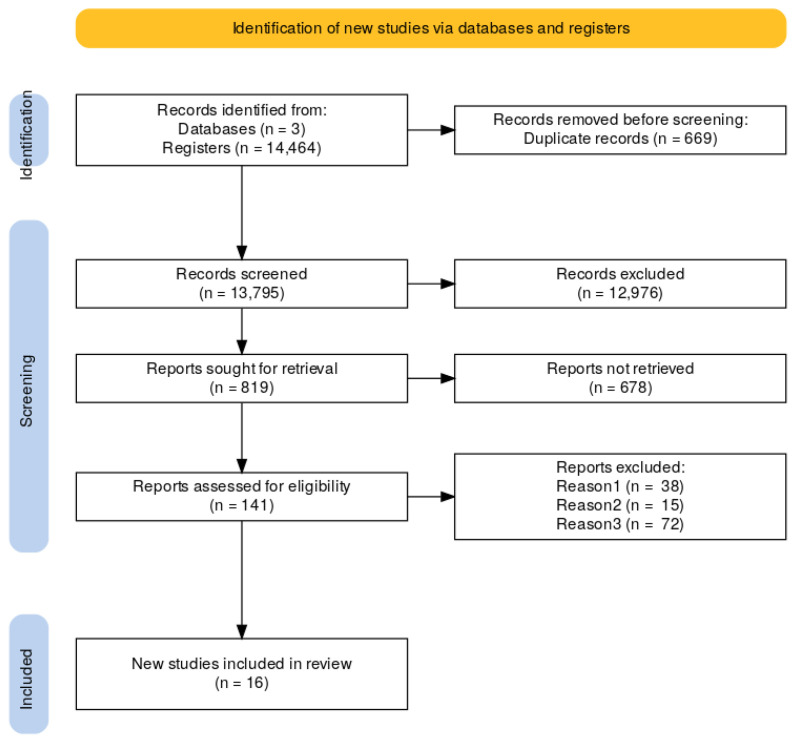
PRISMA diagram.

**Figure 2 brainsci-15-00324-f002:**
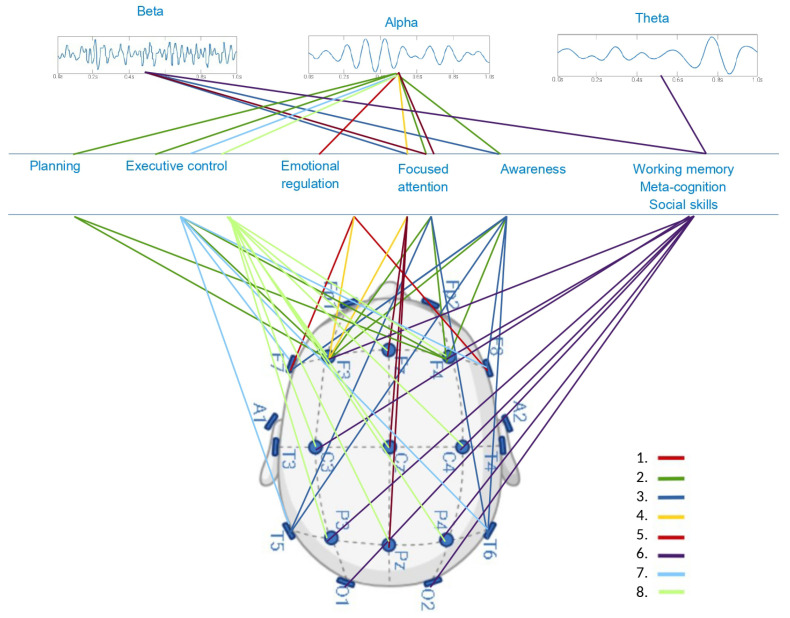
Neuropsychological and electroencephalographical findings showing statistically significant correlation (*p* < 0.001). Lines represent the reported association, and numbers the corresponding reference, 1. Lomas et al., 2015 [55], 2. Mak et al., 2018 [56], 3. Lee et al., 2017 [59], 4. Deng et al., 2021 [61], 5. Morais et al., 2021 [73], 6. Bajestani et al., 2024 [64], 7. Brown et al., 2022 [71], 8. Nien et al., 2023 [69].

**Figure 3 brainsci-15-00324-f003:**
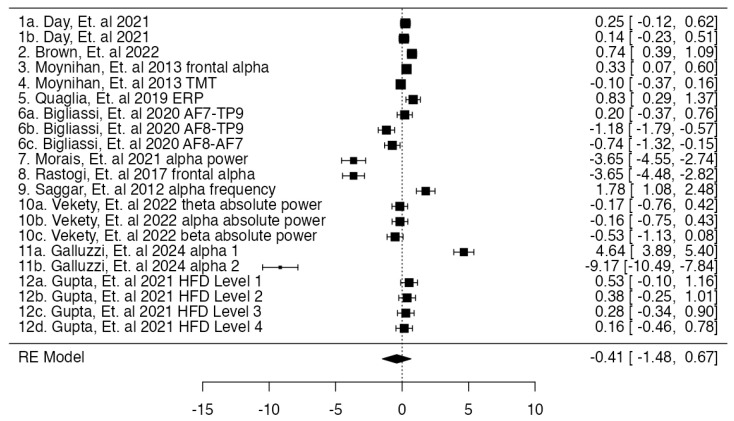
Forest plot. Diamond figure showing mean effect size, 1a, 1b. [60], 2. [71], 3, 4. [72], 5. [75], 6a, 6b, 6c. [68], 7. [73], 8. [74], 9. [63], 10a, 10b, 10c. [70], 11a, 11b. [65], and 12a, 12b, 12c, 12d. [66].

**Figure 4 brainsci-15-00324-f004:**
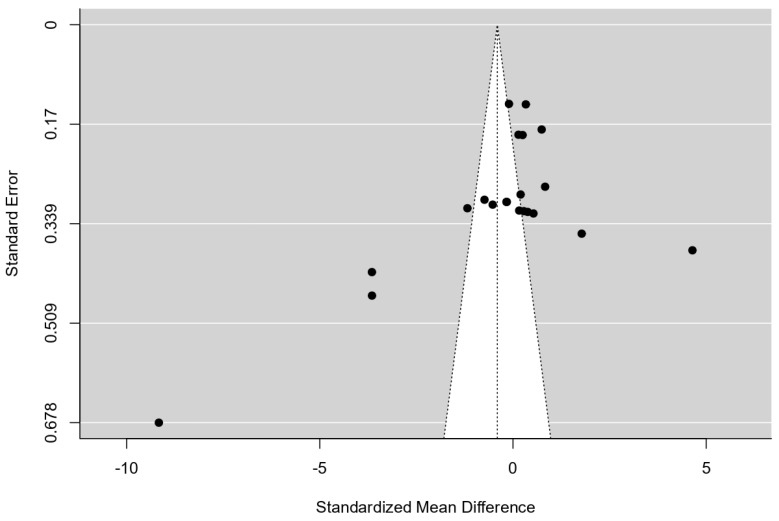
Funnel plot, standard errors, and standardized mean difference.

## Data Availability

Data for this systematic review are available at: DOI: 10.13140/RG.2.2.12075.55849.

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
