# Peer review of "Mindfulness-Based Intervention Effects on EEG and Executive Functions: A Systematic Review"

_brainsci, 2025, doi:10.3390/brainsci15030324_

Round 1
Reviewer 1 Report
Comments and Suggestions for Authors
The authors have done a great job of analyzing a large number of studies. The authors' conclusion is quite clear that even though a large literature reports positive effects of MBI on several behavioral and neurophysiological domains, there are still confounding factors that researchers and clinicians must consider before attributing possible inaccurate or generalizable benefits.
It would have been nice if the authors had addressed several very significant reasons for the disagreement in results.
First, most studies are conducted by inviting women of reproductive age to participate in the study This means that it is simply not possible to average the results of psychological and electrophysiologic studies due to the heterogeneity of these measures in different phases of the menstrual cycle including changing the individual alpha peak frequency and neuronal activation (for ex. Bazanova OM, Nikolenko ED, Barry RJ. Reactivity of alpha rhythms to eyes opening (the Berger effect) during menstrual cycle phases. Int J Psychophysiol. 2017 Dec;122:56-64. doi: 10.1016/j.ijpsycho.2017.05.001) .
Secondly, the disagreement of data on the effect of MBI on electrophysiological parameters is due to the fact that most studies were conducted without taking into account the limits of frequency ranges individually established depending on the alpha peak frequency.
Thirdly, EEG analysis especially in delta and beta bands was performed without taking into account EEG contamination by low-amplitude low-frequency EMG artifacts. Meanwhile, it is the change in EMG that testifies to the change in psychoemotional tension, which is the target of MBI (Armbruster D, Grage T, Kirschbaum C, Strobel A. Processing emotions: Effects of menstrual cycle phase and premenstrual symptoms on the startle reflex, facial EMG and heart rate. Behav Brain Res. 2018 Oct 1;351:178-187. doi: 10.1016/j.bbr.2018.05.030).
If the authors add the above limitations to the discussion chapter, the review will only gain in credibility. In any case, the review deserves to be published.
Author Response
We appreciate the dedication of Reviewer 1 to our manuscript submission. In the review comments, we have found a very interesting approach, analysis, and opportunity to improve, which we now address as follows: for each comment and change made in the manuscript, we marked the text in color blue as well as delivered a brief description in response for each comment.
Comment 1
First, most studies are conducted by inviting women of reproductive age to participate in the study This means that it is simply not possible to average the results of psychological and electrophysiologic studies due to the heterogeneity of these measures in different phases of the menstrual cycle including changing the individual alpha peak frequency and neuronal activation (for ex. Bazanova OM, Nikolenko ED, Barry RJ. Reactivity of alpha rhythms to eyes opening (the Berger effect) during menstrual cycle phases. Int J Psychophysiol. 2017 Dec;122:56-64. doi: 10.1016/j.ijpsycho.2017.05.001).
Response 1
We thank the reviewer for this accurate suggestion regarding the limitations of the studies. Indeed, this is an essential issue regarding heterogeneity. In response to this suggestion, we added sex information from each study in the “Intervention types and gender” column in Table 1 to better describe where most of the data comes from. In addition, we included new discussions related to the suggested references and complementary ones (please refer to lines 603-609). We added this topic as a limitation and challenge of this systematic review for future published data extraction (please refer to lines 621-624).
Comment 2
Secondly, the disagreement of data on the effect of MBI on electrophysiological parameters is due to the fact that most studies were conducted without taking into account the limits of frequency ranges individually established depending on the alpha peak frequency.
Response 2
We thank this very precise observation. Indeed, as mentioned by the reviewer, it may be challenging to generalize EEG measures, considering that EEG feature extraction is extensive. For this study, we described the reported EEF features as major outcomes in Sections 2.4.1, Bandwidth and Region of Interest, 2.4.2, Mean power spectral densities (μV2/Hz), and 2.4.3, Alpha asymmetry. In this Section, we report what the included studies considered for their specific aims (Lines 284-299).
Comment 3
Thirdly, EEG analysis especially in delta and beta bands was performed without taking into account EEG contamination by low-amplitude low-frequency EMG artifacts. Meanwhile, it is the change in EMG that testifies to the change in psychoemotional tension, which is the target of MBI (Armbruster D, Grage T, Kirschbaum C, Strobel A. Processing emotions: Effects of menstrual cycle phase and premenstrual symptoms on the startle reflex, facial EMG and heart rate. Behav Brain Res. 2018 Oct 1;351:178-187. doi: 10.1016/j.bbr.2018.05.030).
Response 3
We agree with this comment. Free of artifact EEG is crucial for adequately making inferences when low-frequency bands are the focus of interest and distinguishing them from wide muscle artifacts or high frequency from specific muscle tension; once quantitatively, these EEG features may be the same, but qualitatively, they keep substantial differences. For this purpose, we added new information from studies reporting the used filtering techniques and signal gathering reliability reported. For example, high and low pass bands, channel interpolation techniques, or independent component analysis in the case of eye movements. Please refer to Section 2.4 Major Outcomes lines 284-299 for this newly added information related to digital and visual examination of the EEG signals.

Reviewer 2 Report
Comments and Suggestions for Authors
The manuscript provides a systematic review of the effect of mindfulness meditation on measures of brain activity (EEG) in their relationship to symptoms of affective impairment and cognitive control capacity. In general, this is a good review on this topic, which presents in detail the results of many experimental studies.
However, I still have a few suggestions for authors that they may want to use to improve the manuscript.
1) I would recommend including in the review's context the studies on the effects of meditation on functional connectivity between different cortical areas. Specifically, the internal default-mode network (DMN) connectivity, as well as connectivity between DMN and other cortical regions is a pattern associated with ruminations and depression, whereas mindfulness meditation significantly affects DMN's connectivity. Perhaps the addition of materials on inter-regional EEG connectivity will improve the manuscript's quality.
2) I also recommend including in the review some EEG studies reporting on the impact of mindfulness meditation on the recognition of self-referential information (recognizing one's own faces, understanding self-related emotional sentences, etc). Violation of self-referential processes is one of the main symptoms of depression, while mindfulness meditation can significantly improve these processes.
3) I recommend in the Introduction section to briefly discuss the functional role of increasing or decreasing alpha and theta rhythm as a marker of emotional or cognitive information processing in the brain. Could you give more detailed review how can we interpret the increase or decrease in spectral power of different rhythms in terms of the involvement of different brain functions in information processing?
Author Response
Reviewer 2
We appreciate the dedication of Reviewer 2 to our manuscript submission. In the review comments, we have found a very interesting approach, analysis, and opportunity to improve, which we now address as follows: for each comment and change made in the manuscript, we marked the text in color blue as well as delivered a brief description in response for each comment.
Comment 1
I would recommend including in the review's context the studies on the effects of meditation on functional connectivity between different cortical areas. Specifically, the internal default-mode network (DMN) connectivity, as well as connectivity between DMN and other cortical regions is a pattern associated with ruminations and depression, whereas mindfulness meditation significantly affects DMN's connectivity. Perhaps the addition of materials on inter-regional EEG connectivity will improve the manuscript's quality.
Response 1
This is an interesting proposal for this systematic review. The suggested topic made by the reviewer was previously approached in the Introduction Section, specifically as one of few research models that propose lower involvement of the DMN at rest, related to a “task readiness.” We kindly suggest referring to lines 59-65. Following this suggestion, we added new research related to the exploration of the DMN using different neuroimaging techniques.
Comment 2
I also recommend including in the review some EEG studies reporting on the impact of mindfulness meditation on the recognition of self-referential information (recognizing one's own faces, understanding self-related emotional sentences, etc). Violation of self-referential processes is one of the main symptoms of depression, while mindfulness meditation can significantly improve these processes.
Response 2
We thank the reviewer for this accurate suggestion. It is an excellent part of the present work regarding the possible effects of meditation on social cognition skills. From this point of view, self-referential is a relevant source of information for decision-making in social environments. Please refer to the newly added findings reported in lines 221-223.
Comment 3
I recommend in the Introduction section to briefly discuss the functional role of increasing or decreasing alpha and theta rhythm as a marker of emotional or cognitive information processing in the brain. Could you give more detailed review how can we interpret the increase or decrease in spectral power of different rhythms in terms of the involvement of different brain functions in information processing?
Response 3
This comment certainly helps the manuscript improve the consistency and reliability of the framework proposals. We added a new background related to EEG alpha rhythm and its relation with emotional regulation. Please refer to lines 120-139, where we present historically well-described findings related to specific alpha and theta changes and their relation with emotional states or mood conditions. Given the suggested topic's relevance, we added new findings from the literature addressing further details on the above-mentioned frequency modifications.
